# Rethinking Audiovisual Segmentation with Semantic Quantization and Decomposition

## Abstract

Audiovisual segmentation (AVS) is a challenging task that aims to segment visual objects in videos based on their associated acoustic cues. With multiple sound sources involved, establishing robust correspondences between audio and visual contents poses unique challenges due to its (1) intricate entanglement across sound sources and (2) frequent shift among sound events. Assuming sound events occur independently, the multi-source semantic space (which encompasses all possible semantic categories) can be represented as the Cartesian product of single-source sub-spaces. This motivates us to decompose the multi-source audio semantics into single-source semantics, enabling more effective interaction with visual content. Specifically, we propose a semantic decomposition method based on product quantization, where the multi-source semantics can be decomposed and represented by several quantized single-source semantics. Furthermore, we introduce a global-to-local quantization mechanism, which distills knowledge from stable global (clip-level) features into local (frame-level) ones, to handle the constant shift of audio semantics. Extensive experiments demonstrate that semantically quantized and decomposed audio representation significantly improves AVS performance, e.g., +21.2% mIoU on the most challenging AVS-Semantic benchmark.

## 1 Introduction

Recently, audiovisual segmentation (AVS) (Zhou et al., 2022) is introduced to explore audiovisual correlations at the pixel level. Specifically, AVS aims to segment sounding object(s) in video frames with the associated audio. Audiovisual semantic segmentation (AVSS) (Zhou et al., 2023) extends AVS by additionally identifying the categories of sound sources. As shown in Fig. 1 (a), in contrast to the visual domain, where each pixel has a unique semantic label, multi-source audio is temporally entangled, leading to ambiguity when associating the hybrid audio with semantically distinct visual contents. This motivates us to explore suitable representations of multi-source audio for more effective audiovisual interactions.

Let us commence by exploring the simplest scenario (left of Fig. 1 (a)), involving a single sound source. In this scenario, each visual pixel or acoustic timestep is only associated with one semantic label. Here, we denote the set containing all possible semantic labels for pixels as visual semantic

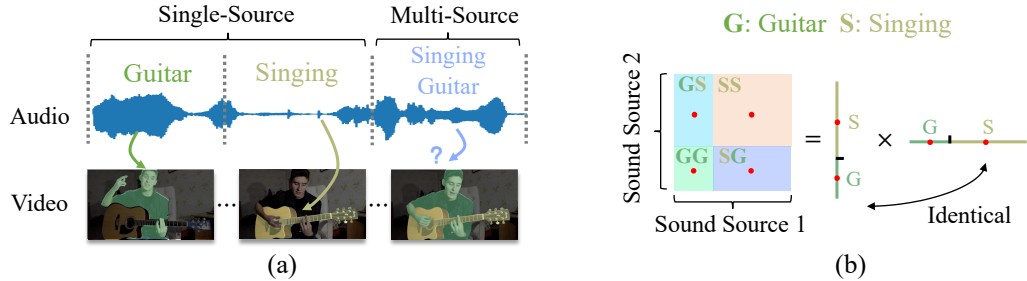

Figure 1: (a) **Audiovisual semantic interaction.** (b) **Semantic decomposition.** Multi-source audio semantic space can be assumed as a Cartesian product of single-source semantics, which can be decomposed via product quantization. The red points represent the quantized semantics.

space and those for acoustic timesteps as acoustic semantic space, respectively. In this example, we can find that both visual and acoustic single-source semantic spaces share the same semantic labels as {Guitar (G), Singing (S)}. Let us consider the two-source moment (right of Fig. 1 (a)). The visual semantics remain the same as in previous frames, but the size of possible two-source audio semantics presents a quadratic increase ({GG, GS, SG, SS}). This not only increases the difficulty in modeling larger semantic spaces but also complicates the alignment between visual and acoustic semantic spaces. The complexity further intensifies as more sources come into play.

Unlike previous methods (Zhou et al., 2023; 2022) that directly interact entangled multi-source audio representation with visual contents, we intend to disentangle the multi-source audio semantics into several single-source semantics for further more effective audiovisual interaction. We simplify the problem by assuming independent sound events, which allows us to represent the multi-source semantic space as a Cartesian product of identical single-source semantic spaces. In specific, we introduce a product quantization-based (PQ-based) method to decompose the multi-source semantics. Product quantization aims to represent a complex space through the product of several subspaces. In the multi-source case, single-source semantics can serve as subspaces. We show the semantic decomposition of a two-source example in Fig. 1 (b) where the single-source semantic subspaces share identical semantics {G, S}. Specifically, product quantization can be easily achieved by learning separate transforms of multi-source semantics and then quantizing them utilizing a shared codebook as shown in Fig. 2. We interact the decomposed single-source semantics with visual features for effective alignment.

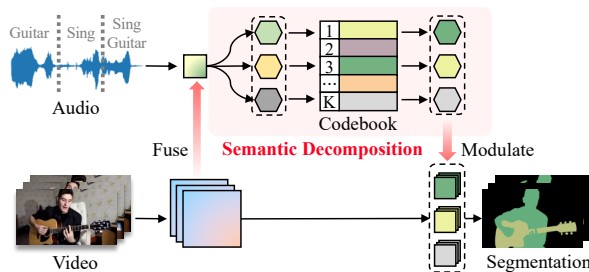

Figure 2: Semantic decomposition via product quantization (with sharing codebook for subspaces).

Furthermore, considering that active sound events may continually change over time, another challenge for AVS is to extract frame-level audio semantics which is typically not as robust as extracting from clip-level audio. To improve the frame-level audio representation, we propose a global-to-local mechanism, which distills knowledge from robust global (clip-level) audio representations into local (frame-level) ones. Specifically, we build an effective codebook for semantic quantization with clip-level visual-enriched audio features and then apply this codebook to perform local quantization on each frame without updating it. Thereby, the local semantic tokens are calibrated to the more robust and representative clip-level feature in the codebook.

In summary, our contribution is three-fold:

- An effective approach of multi-source audio semantic decomposition via product quantization, addressing the challenge of interacting visual and audio features in multiple object scenarios.

- A global-to-local distilling mechanism for frame-level audio semantic enhancement, addressing the ineffectiveness of frame-level audio feature extraction.

- Extensive experiments are conducted to verify the effectiveness of the proposed method, which significantly outperforms previous state-of-the-art methods on three AVS benchmarks, especially for multi-object datasets (+5.4% mIoU for AVS-Objec-Multi and +21.2% mIoU for AVS-Semantic).

## 2 RELATED WORK

**Audiovisual segmentation and localization.** Audiovisual segmentation (AVS), which was recently introduced (Zhou et al., 2022), aims to segment the objects that produce sound at the time of the image frame. Zhou *et al.* (Zhou et al., 2022) proposed a method with cross-modal attention to locate the sound source, making it the pioneering work in AVS. Recently, an extended task of AVS, audiovisual semantic segmentation (AVSS), is proposed by Zhou *et al.* (Zhou et al., 2023) which aims to not only segment the mask of sound sources but also predict the category of each sound

source. Due to the semantic entanglement in audio, tackling multi-source AVSS is more challenging than AVS task. Zhou *et al.* (Zhou et al., 2023) follows the TPAVI module in (Zhou et al., 2022) to conduct audiovisual interaction. Sound source localization (SSL) (Mo & Morgado, 2022a;b; Senocak et al., 2018; Hu et al., 2019; Qian et al., 2020; Chen et al., 2021; Afouras et al., 2020) is a related problem to AVS that aims to locate the regions of sounds in the visual frame. Common SSL methods (Arandjelovic & Zisserman, 2018; 2017; Cheng et al., 2020; Senocak et al., 2018) leverage cross-modal correspondence between audio and visual features to locate sounds, which are then displayed as heatmaps. For instance, Mo *et al.* (Mo & Morgado, 2022a) leverage multi-level audiovisual contrastive learning to effectively locate the objects. Different from previous methods primarily designed for single-source scenarios, our objective is to address the semantic entanglement present in multi-source audios and explore methods for effective interaction between multi-source audios and videos.

**Audio-visual learning.** Audio-visual learning has been explored in many works (Aytar et al., 2016; Arandjelovic & Zisserman, 2017; Korbar et al., 2018; Senocak et al., 2018; Zhao et al., 2018; 2019; Gan et al., 2020; Georgescu et al., 2022) which aims to learn audio-visual correspondence from paired audio-visual data. Most methods maximize the mutual information between corresponding audio and video pairs by several proxy tasks. Constructing negative samples (Zhao et al., 2018; 2019; Gan et al., 2020) and learning to push them away while closing positive ones is a common goal. Recently, another track (Georgescu et al., 2022; Gong et al., 2022) masks information in audio-visual pairs and tries to reconstruct incomplete information in one modality by conditioning on the other. The learned correspondence can be leveraged for several tasks, such as audio-visual source localization (Mo & Morgado, 2022a;b; Senocak et al., 2018; Hu et al., 2019; Qian et al., 2020; Chen et al., 2021; Afouras et al., 2020), audio-visual separation (Gao & Grauman, 2019; Morgado et al., 2018; 2020; Chen et al., 2020a), audio-visual parsing (Wu & Yang, 2021; Mo & Tian, 2022; Lin et al., 2021; Tian et al., 2020). In this work, we focus on how to effectively construct correspondence between multi-source audio and video for fine-grained audiovisual segmentation which is more challenging due to the entanglement of semantics in audio.

# 3 METHOD

In this section, we first present the formulation of the product quantization-based (PQ-based) method for multi-source audio semantic decomposition. Then, we outline the pipeline that utilizes the quantized and decomposed audio representation to improve the audiovisual segmentation tasks.

## 3.1 PQ-BASED MULTI-SOURCE SEMANTIC DECOMPOSITION.

The core of the PQ-based decomposition is to concisely represent the multi-source semantic space $\mathcal{X}_m$ with the product of multiple single-source semantic spaces $\mathcal{X}_s$.

Given a codebook containing a finite set of codewords $\mathcal{C} = \{e_i\}_{i=1}^K$, the vector quantizer $\mathrm{VQ}(\cdot)$ maps a feature $x \in \mathcal{X}$ to a codeword $e_i = \arg\min_{e_i \in \mathcal{C}} \|x - e_i\|_p$ that minimizes the distance between $x$ and $e_i$ in the $p$-norm sense. As the single-source audio is semantically unique for each time step, for single-source space with $K$ sound event categories, a codebook $C_s$ with $K_s = K$ codewords can sufficiently encode the space $\mathcal{X}_s$ without losing information. Nevertheless, for a $N$-source semantic space, a combination of sound events can appear for each time step. Therein, to fully represent the space, a codebook $\mathcal{C}_m$ of size $K_m = K^N$ is required.

We assume a $N$-source semantic space $\mathcal{X}_m$ is a Cartesian product of several identical single-source semantic spaces $\mathcal{X}_s$ as

$$\mathcal{X}_m = \underbrace{\mathcal{X}_s \times \cdots \times \mathcal{X}_s}_{N}. \tag{1}$$

Specifically, we can obtain the product quantization of $x \in \mathcal{X}_m$ through an order-invariant operation, *i.e.*, concatenation, on separately quantized $x_i = f_i(x)$, where $f_i(\cdot)$ is a transform applied on $x$:

$$\mathrm{PQ}(x) = \mathrm{VQ}_s(x_1) \oplus \cdots \oplus \mathrm{VQ}_s(x_N), \quad \text{w.r.t } \mathrm{VQ}_s \sim \mathcal{C}_s. \tag{2}$$

$\mathrm{VQ}_s \sim \mathcal{C}_s$ denotes the $\mathrm{VQ}_s(\cdot)$ is associated with codebook $\mathcal{C}_n$ and $\oplus$ denotes channel-wise concatenation. As the codebook $\mathcal{C}_s$ is shared with all $\mathrm{VQ}_s(\cdot)$, the codebook for the multi-source semantic

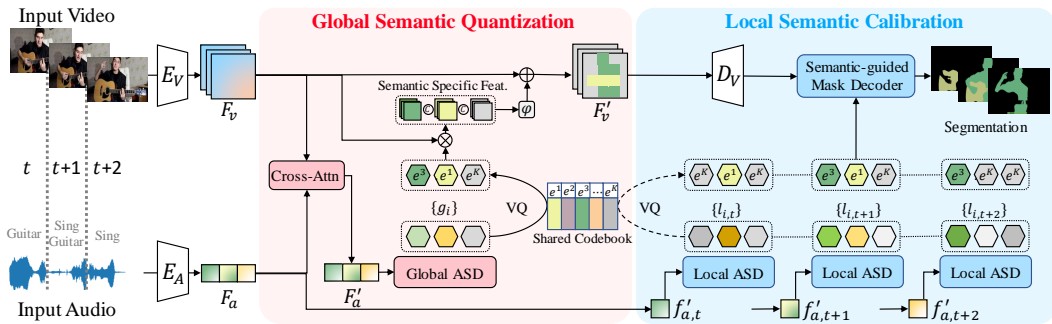

Figure 3: Method overview. The **global semantic quantization** module decomposes multi-source audio features and enables interaction between the decomposed single-source audio feature and visual features. The **local semantic calibration** module distills knowledge from global (clip-level) audio features to local (frame-level) audio features by utilizing a shared codebook, which stores quantized audio representation during semantic quantization.

space $\mathcal{C}_m$ is reduced to be the same as $\mathcal{C}_s$. By constraining the size of the single-source codebook $K_s \ll K^N$, we can force the transform $f_i(\cdot)$ to decompose the multi-source semantics $x$.

## 3.2 NETWORK OVERVIEW

We present the proposed framework with **S**emantically **Q**uantized and **D**ecomposed (SQD) audio representation consisting of three main components: feature encoding, global semantic quantization and local semantic calibration, as illustrated in Fig. 3.

(1) First, we extract visual features $F_v = \{f_{v,t}\}_{t=1}^{T}$ and acoustic features $F_a = \{f_{a,t}\}_{t=1}^{T}$ by separate encoders. (2) Then, **to decompose semantics in multi-source audio features**, we use a global semantic decomposition module to map the audio query into a set of semantic tokens $\{g_i\}_{i=1}^{N}$. We then learn a semantic codebook to quantize them. The quantized tokens are further employed to modulate the visual features to inject information about corresponding sound sources. (3) Afterwards, **to obtain frame-level audio features to query object masks**, we utilize a local semantic decomposition module for each time step, which uses the global codebook to decouple local audio semantics. Each quantized local semantic token $\text{VQ}(l_{i,t})$ serves as a query to segment a frame-level mask with the semantic-guided mask decoder. Overall, the proposed SQD boosts AVS by enhancing the audiovisual semantic interaction.

## 3.3 GLOBAL SEMANTIC QUANTIZATION

To tackle the mixture of multi-source audio queries and effectively conduct audiovisual fusion, we propose global semantic quantization to decompose audio semantics, which consists of two steps: global semantic decomposition and audiovisual semantic recombination. The detailed structure of the modules is illustrated in Fig. 4.

**Global semantic decomposition.** Global semantic decomposition aims to decompose multi-source audio semantics into single-source semantics. Specifically, the audio feature $F_a$ is first fused with video feature $F_v$ to be $F_a'$, taking the form:

$$F_a' = \text{LN}(\text{FFN}(h_a) + h_a),$$
$$h_a = \text{LN}(\text{MCA}(F_a, F_v) + F_a), \tag{3}$$

where MCA denotes Multi-head Cross-Attention, LN denotes Layer Normalization, and FFN denotes Feed-Fordward Network. After that, we transform the audio feature $F_a'$ to $N$ decomposed semantic tokens $\{g_i\}_{i=1}^{N}$ with a global audio semantic decoder (global ASD):

$$g_i = \text{TrD}_{global}(p_i|F_a') \tag{4}$$

by querying a set of learnable semantic prototypes $\{p_i\}_{i=1}^{N}$ to the feature $F_a$, with a transformer decoder $\text{TrD}_{global}$. Each semantic token is then quantized to be $e_i = \text{VQ}(g_i)$ with the shared codebook $\mathcal{C} = \{e^k\}_{k=1}^{K}$, imposing that all semantic tokens to share an identical feature subspace

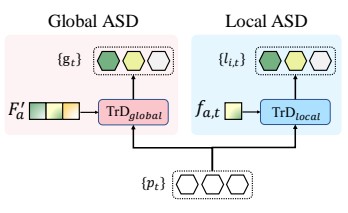 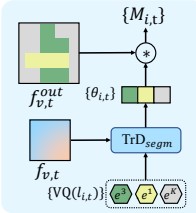

(a) Audio Semantic Decoder      (b) Semantic-guided Mask Decoder

Figure 4: (a) Global and local audio semantic decoder (ASD) share similar structures that query clip-/frame-level audio features, $F'_a$ or $f_{a,t}$, with a transformer decoder $\mathrm{TrD}_{global}/\mathrm{TrD}_{local}$ using learnable semantic prototypes $\{p_i\}$. (b) The semantic-guided mask decoder contains a transformer decoder $\mathrm{TrD}_{segm}$ to align audiovisual features and computes dynamic filters $\theta_{i,t}$. The final mask $M_{i,t}$ is generated by a dynamic convolution between the visual feature $f_{v,t}^{out}$ and $\theta_{i,t}$.

with low cardinality. Note that we set the codebook size $K \ll D_{semantic}^N$ to force the network to learn decomposed semantics, where $D_{semantic}^N$ is the number of sound event categories $D_{semantic}$ to the power of $N$.

**Audiovisual semantic recombination.** Audiovisual semantic recombination aims to leverage the decomposed audio feature to interact with visual features. After obtaining quantized global semantic tokens $\{e_i\}_{i=1}^N$, which encode $N$ groups of decomposed semantics, we aim to interact them with visual features while preserving the original function of the multi-source audio input. A set of dynamic filters $\{w_i \in \mathbb{R}^{C_v}\}_{i=1}^N$ are first learned from global semantic tokens $\{g_i\}_{i=1}^N$ by two linear layers. After that, we utilize channel-wise attention to modulate video features by each filter to interact the visual feature with the content referred by different semantic tokens, which is given by:

$$F'_v = \mathrm{BN}(\varphi(w_i F_v \oplus \cdots \oplus w_N F_v) + F_v), \tag{5}$$

where $\varphi$ denotes a convolution layer to reduce channel from $N \times C_v$ to $C_v$, BN denotes Batch Normalization, and $\oplus$ denotes concatenation among channels. By incorporating channel-wise attention, the visual features can be more effectively concentrated on the relevant audio content. Furthermore, through channel-wise concatenation, the decomposed audio semantics can be reintegrated, producing hybrid semantics that refers to the holistic contents of the original audio input.

### 3.4 LOCAL SEMANTIC CALIBRATION

Since the audio query is time-variant, global semantic tokens cannot be accurately aligned with visual features at the frame level. To segment audio-queried contents in each frame, we propose the local semantic calibration, consisting of a local semantic decomposition stage and a semantic-guided mask decoding stage.

**Local semantic decomposition.** Local semantic decomposition module aims to decompose the semantics encoded in each audio frame. Similar to the global semantic decoder, the local semantic decoder (Local ASD) decodes frame-level semantics with a transformer decoder $\mathrm{TrD}_{local}$ and a set of semantic prototypes $\{p_i\}_{i=1}^N$. The local semantic tokens $l_{i,t}$ are given by

$$l_{i,t} = \mathrm{TrD}_{local}(p_i | f_{a,t}). \tag{6}$$

The local semantic tokens do not build their own codebook but utilize the global codebook $\mathcal{C}$, that is, they do not update $\mathcal{C}$ but are committed to being close to the vectors in $\mathcal{C}$. In this way, the local semantic tokens distill knowledge from the global ones. Further explanation regarding supervision will be provided in Section 3.5.

**Semantic-guided mask decoding.** We utilize the semantic-guided mask decoder to decode visual features into masks that correspond to decomposed local audio semantics, with detailed structure illustrated in Fig. 4 (b). Pyramid video features $F_v^{out} = \{f_{v,t}^{out}\}_{t=1}^T$ are obtained with the feature pyramid network (Lin et al., 2017a). We leverage a shared multimodal transformer decoder $\mathrm{TrD}_{segm}$ to generate dynamic filters $\theta_{i,t} = \phi_{segm}(\mathrm{TrD}_{segm}(\mathrm{VQ}(l_{i,t})|f_t))$ for each timestep, where $\phi_{segm}$ is a two-layer fully-connected network. The final mask segmentation can be obtained by:

$$M_{i,t} = f_t^{out} * \theta_{i,t}, \tag{7}$$

| Method | Backbone | AVS-Object-Single | | | AVS-Object-Multi | | | AVS-Sementic |
|--------|----------|-------------------|--|--|------------------|--|--|--------------|
| | | $\mathcal{J}\&\mathcal{F}\uparrow$ | $\mathcal{J}\uparrow$ | $\mathcal{F}\uparrow$ | $\mathcal{J}\&\mathcal{F}\uparrow$ | $\mathcal{J}\uparrow$ | $\mathcal{F}\uparrow$ | mIoU$\uparrow$ |
| ResNet Backbone | | | | | | | | |
| LVS (Chen et al., 2021) | ResNet-18 | 44.5 | 37.9 | 51.9 | 31.3 | 29.5 | 33.0 | - |
| MSSL (Qian et al., 2020) | ResNet-18 | 55.6 | 44.9 | 66.3 | 31.4 | 26.1 | 36.3 | - |
| 3DC (Mahadevan et al., 2020) | 3DC | 66.5 | 57.1 | 75.9 | 43.6 | 36.9 | 50.3 | 17.3 |
| AOT (Yang et al., 2021) | ResNet-50 | - | - | - | - | - | - | 25.4 |
| AVS (Zhou et al., 2023) | ResNet-50 | 78.8 | 72.8 | 84.8 | 53.6 | 47.9 | 57.8 | 20.2 |
| Bi-Gen (Hao et al., 2023) | ResNet-50 | 79.8 | 74.1 | 85.4 | 50.9 | 50.0 | 56.8 | - |
| AVSegFormer (Gao et al., 2023) | ResNet-50 | 81.2 | 76.5 | 85.9 | 56.2 | 49.5 | 62.8 | 24.9 |
| **SQD (Ours)** | ResNet-50 | **81.8** | **77.6** | **86.0** | **61.6** | **59.6** | **63.5** | **46.6** |
| Transformer Backbone | | | | | | | | |
| iGAN (Mao et al., 2021) | Swin-Base* | 69.7 | 61.6 | 77.8 | 48.7 | 42.9 | 54.4 | - |
| SST (Duke et al., 2021) | SSL | 73.2 | 66.3 | 80.1 | 49.9 | 42.6 | 57.2 | - |
| LGVT (Zhang et al., 2021) | Swin-Base* | 81.1 | 74.9 | 87.3 | 50.0 | 40.7 | 59.3 | - |
| AVS (Zhou et al., 2023) | PVT-v2-Base | 83.3 | 78.7 | 87.9 | 59.3 | 54.0 | 64.5 | 29.8 |
| **SQD (Ours)** | Swin-Tiny | **83.9** | **79.5** | **88.2** | **64.0** | **61.9** | **66.1** | **53.4** |
| **SQD (Ours)** | V-Swin-Tiny | **84.7** | **80.7** | **88.7** | **65.4** | **63.7** | **67.0** | **54.7** |

Table 1: **Quantitative comparison to AVS and AVSS methods.** Swin-Base* denotes modified Swin-Base Transformer (Liu et al., 2021). SSL is Sparse Spatiotemporal Transformers (Duke et al., 2021). PVT-v2 (Wang et al., 2022) is a strong Pyramid Vision Transformer. V-Swin-Tiny is the Video Swin Transformer (Liu et al., 2022). $\uparrow$ indicates the larger the better.

where $*$ denotes the dynamic convolution (Chen et al., 2020b). Each filter represents semantics of a decomposed single-source audio, contributing to the segmentation of the single sounding object. Additional class probability prediction $P_{i,t}$ and bounding box prediction $B_{i,t}$ for each mask $M_{i,t}$ are performed by two two-layer fully connected networks from the output of $\text{TrD}_{segm}(\text{VQ}(l_{i,t})|f_t)$.

### 3.5 Loss Function

The overall loss function is given by

$$\mathcal{L} = \lambda_{quant}\mathcal{L}_{quant} + \mathcal{L}_{segm}, \tag{8}$$

where $\mathcal{L}_{quant}$ and $\mathcal{L}_{segm}$ are the loss for semantic quantization and semantic segmentation, respectively. $\lambda_{quant}$ is a constant.

**Loss for semantic quantization.** The quantizer is shared with both global and local semantic decomposition, while the local semantic tokens do not update the codebook. The loss is given by

$$\mathcal{L}_{quant} = \sum_{i=1}^{N} \|\text{VQ}(g_i) - \text{sg}[g_i]\|_2^2$$
$$+ \lambda_{com}\|\text{sg}[\text{VQ}(g_i)] - g_i\|_2^2 + \lambda_{com}\|\text{sg}[\text{VQ}(l_i)] - l_i\|_2^2, \tag{9}$$

where $\text{sg}[\cdot]$ stands for stop-gradient operation. $\text{VQ}(\cdot)$ denotes the vector quantization function, where $\text{VQ}(x) = e_i = \arg\min_{e_i} \|x - e_i\|_2 \in \mathcal{C}$ and $\mathcal{C} = \{e_i\}_{i=1}^{K}$ is the shared codebook. The first term aims to update the codebook. The second and third terms aim to minimize the quantization error by forcing the input vector to be quantized to its closest vector in the codebook.

**Loss for semantic segmentation.** Let the predictions of the network be $\mathbf{y} = \{\mathbf{y}_i\}_{i=1}^{N}$ where $y_i = \{B_{i,t}, P_{i,t}, M_{i,t}\}_{t=1}^{T}$. $B_{i,t}$, $P_{i,t}$ and $M_{i,t}$ denote bounding box, class probability and mask predictions respectively. We denote the ground-truth as $\hat{\mathbf{y}} = \{\hat{\mathbf{y}}_j\}_{j=1}^{N}$ (padded with $\emptyset$ (Cheng et al., 2021a)) where $\hat{y}_j = \{\hat{B}_{j,t}, \hat{C}_{j,t}, \hat{M}_{j,t}\}_{t=1}^{T}$. $C_{j,t}$ is the ground-truth class for the $j$-th sounding object in the video at $t$ frame. We search for an assignment $\sigma \in \mathcal{P}_N$ with the highest similarity where $\mathcal{P}_N$ is a set of permutations of N elements . The similarity can be computed as

$$\mathcal{L}_{match}(y_i, \hat{y}_j) = \lambda_{box}\mathcal{L}_{box} + \lambda_{cls}\mathcal{L}_{cls} + \lambda_{mask}\mathcal{L}_{mask}, \tag{10}$$

where $\lambda_{box}$, $\lambda_{cls}$, and $\lambda_{mask}$ are weights to balance losses. We leverage a combination of Dice (Li et al., 2019) and BCE loss as $\mathcal{L}_{mask}$, focal loss (Lin et al., 2017b) as $\mathcal{L}_{cls}$, and GIoU (Rezatofighi et al., 2019) and L1 loss as $\mathcal{L}_{box}$. The best assignment $\hat{\sigma}$ is solved by the Hungarian algorithm (Kuhn, 1955). Given the best assignment $\hat{\sigma}$, the segmentation loss between ground-truth and predictions is defined as $\mathcal{L}_{segm} = \mathcal{L}_{match}(y_i, \hat{y}_{\hat{\sigma}(j)})$.

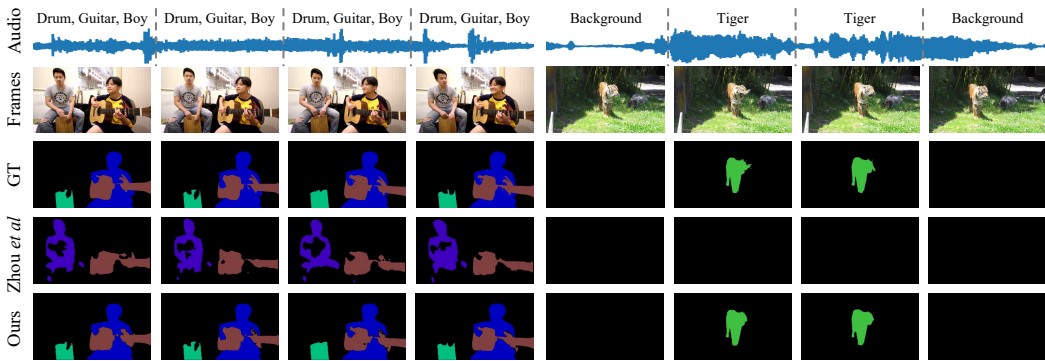

Figure 5: Qualitative comparison to Zhou et al. (Zhou et al., 2023) on AVS-Semantic. Each color represents a semantic category. Note that the class labels in the first row serve as references but are not given in the input.

## 4 EXPERIMENTS

**Dataset.** We conduct experiments on AVS-Object (Zhou et al., 2022) for AVS task and AVS-Semantic (Zhou et al., 2023) for AVSS task.

- **AVS-Object**: AVS-Object dataset contains 5,356 short videos with corresponding audios in which 4,932 audios contain single-source and 424 audios contain multiple sources. Class-agnostic masks are given as annotations for AVS task. Typically, it is evaluated separately for single- and multi-source audios as AVS-Object-Single and AVS-Object-Multi.
- **AVS-Semantic**: AVS-Semantic is an extended dataset from AVS-Object which contains 12,356 videos with 70 classes. Semantic segmentation is annotated for AVSS task. Both single- and multi-source audio cases exist in the AVS-Semantic.

**Metrics.** For AVS task, the convention is to compute region similarity $\mathcal{J}$ and contour accuracy $\mathcal{F}$ as defined in (Pont-Tuset et al., 2017). Note that we follow the video segmentation convention to use the region similarity $\mathcal{J}$, which is equivalent to mIoU in the binary AVS setting. For AVSS, we follow the semantic segmentation convention to evaluate the model using mIoU which is defined as the intersection over union averaged among all classes.

**Implementation Detail.** We implement our method in PyTorch (Paszke et al., 2019). We train our model for 13 epochs and 16 epochs with a learning rate multiplier of 0.1 at the $11^{th}$ and $14^{th}$ epochs for AVS-Object and AVS-Semantic datasets, respectively. The initial learning rate is 1e-4, and a learning rate multiplier of 0.5 is applied to the backbone. We adopt $\mathrm{batchsize} = 4$ and an AdamW (Loshchilov & Hutter, 2017) optimizer with weight decay $5 \times 10^{-4}$. Multi-scale training is adopted to obtain a strong baseline, and if no specification, all images are resized to have the longest side 224 during evaluation. More details are available in the supplementary materials.

### 4.1 MAIN RESULTS

**Quantitative comparison on AVS-Object.** Our method outperforms the previous state-of-the-art (SOTA) method AVSegFormer (Gao et al., 2023) by 0.6 and 5.4 of $\mathcal{J}\&\mathcal{F}$ score on AVS-Object-Single and AVS-Object-Multi datasets respectively (with ResNet-50 backbone). We notice that the improvement on the multi-source setting is much larger than the single-source setting. This is because single-source audios contain simple and disentangled semantics and can be easily aligned with visual features while, for multi-source audios, the complex semantic space makes the alignment to visual contents much more difficult.

**Quantitative comparison on AVS-Semantic.** Compared to the AVS-Object task, our method demonstrates greater improvement in the AVS-Semantic task. As shown in the Table 1, our method eclipses the previous SOTA AVSS method AOT (Yang et al., 2021) by a remarkable 21.2 mIoU with ResNet-50 backbone. The improvement in the AVSS task can be attributed to several factors. First, the task itself involves the semantic prediction of sound sources. However, due to mixed audio

| Module | AVS-Object-Multi | | | AVS-Sementic |
|---|---|---|---|---|
| | $\mathcal{J}\&\mathcal{F}\uparrow$ | $\mathcal{J}\uparrow$ | $\mathcal{F}\uparrow$ | mIoU$\uparrow$ |
| Baseline | 52.9 | 50.1 | 55.7 | 33.5 |
| +GSD | 56.7 | 54.5 | 58.8 | 38.4 |
| +GSD+AVSR | 58.6 | 56.5 | 60.6 | 40.9 |
| +GSD+AVSR+LSD | 60.1 | 58.2 | 61.9 | 44.5 |
| +GSD+AVSR+LSD+SC | 61.6 | 59.6 | 63.5 | 46.6 |

Table 2: Component analysis. GSD: global semantic decomposition; AVSR: audiovisual semantic recombination; LSD: local semantic decomposition; SC: sharing codebook.

| Codebook Size | Object-M $\mathcal{J}\&\mathcal{F}\uparrow$ | Sementic mIoU$\uparrow$ |
|---|---|---|
| 1 | 52.7 | 24.8 |
| 32 | 61.4 | 31.5 |
| 64 | 60.0 | 43.2 |
| 128 | 61.6 | 46.6 |
| 256 | 60.6 | 46.1 |

Table 3: Ablation on codebook size.

| Token Number | Object-M $\mathcal{J}\&\mathcal{F}\uparrow$ | Sementic mIoU$\uparrow$ |
|---|---|---|
| 1 | 59.7 | 40.2 |
| 3 | 61.0 | 43.5 |
| 5 | 61.6 | 46.6 |
| 7 | 61.6 | 45.9 |
| 9 | 61.2 | 46.3 |

Table 4: Ablation on decomposed token number.

| Decomp. Domain | Object-M $\mathcal{J}\&\mathcal{F}\uparrow$ | Sementic mIoU$\uparrow$ |
|---|---|---|
| Time | 56.2 | 38.9 |
| Semantic | 61.6 | 46.6 |

Table 5: Ablation on sound decomposition domain.

signals, aligning visual content accurately becomes challenging, leading to difficulties in classification. Secondly, the number of sound sources and categories of AVS-Semantic dataset are larger than AVS-Object, which will result in a larger complex semantic space. When the mixed semantics are not decomposed, the network struggles to handle the numerous mixed semantics effectively. Thirdly, in the AVS-Semantic dataset, sound event changes occur more frequently. As a result, a more robust frame-level audiovisual correspondence is required. Our proposed global-to-local distilling mechanism addresses this challenge by enhancing the capture of local semantic information, enabling accurate object segmentation.

**Qualitative comparison.** As shown in Fig. 5, we qualitatively compare our method to the method proposed by Zhou et al. (Zhou et al., 2022) on AVS-Semantic. Our method achieves better results on both segmenting quality and class prediction accuracy. Since the method (Zhou et al., 2022) directly fuses mixed audio features with video features, we notice that it suffers from object incorrectness when multiple sound sources are present. Meanwhile, due to the lack of frame-level audiovisual calibration, (Zhou et al., 2022) cannot effectively handle the audio semantic changes. More qualitative results on the AVS-Object dataset are available in the Appendix.

## 4.2 ABLATION STUDY

**Module Effectiveness.** We conduct experiments to validate the effectiveness of our proposed modules. We first construct a **baseline** with unimodal encoders and the semantic-guided mask decoder, and then add other modules step-by-step. As shown in Table 2, each of the proposed modules benefits the performance. For AVS-Semantic, both global semantic decomposition (GSD) and local semantic decomposition (LSD) bring obvious gains; for AVS-object, the LSD only slightly improves the performance. This could be attributed to the longer duration of videos in AVS-Semantic compared to AVS-Object, which allows for a greater number of semantic changes within each clip. Finally, with all components, our method achieves the gains of 10.5 $\mathcal{J}\&\mathcal{F}$ and 13.1 mIoU on AVS-Object-Multi and AVS-Semantic respectively when compared to the baseline.

**Semantic token number.** We ablate the semantic token number for the global-ASD and local-ASD in Table 4. We observe that a token number of 5 yielded the best performance. This can be attributed to the fact that the maximum number of mixed sound sources for the audio-visual dataset is 5.

**Codebook size.** The cardinality of the codebook is essential to our semantic decomposition. Ideally, we aim to constrain the cardinality of the codebook to be close to the semantic category number. We ablate on codebook size from 1 to 256. When the codebook size equals 1, all the decomposed audio tokens are the same, resulting in all the same segmentation results. As shown in Table 3, we notice even a codebook size of 1 achieves 24.8 mIoU on AVS-Semantic. A codebook of size=128 achieves the best performance. Please note that a codebook size slightly larger than the category number, e.g.128, will not hamper the semantic decomposition capability of our method, since $128 \ll 70^N$ where $N > 1$ is the maximum sound source number, and 70 is the category number.

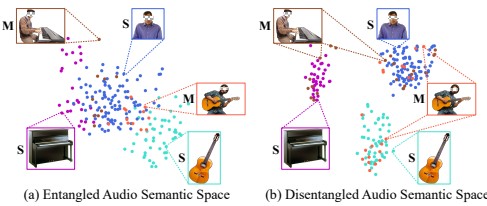

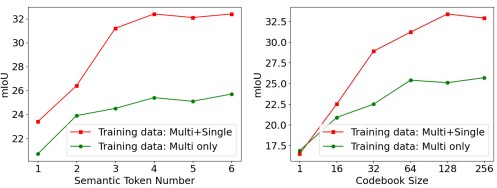

Figure 6: Visualization comparison between entangled and our disentangled audio semantic space. "M" and "S" notations denote multi-source and single-source inputs.

Figure 7: Comparison of training w. and w./o. single-source data.

### 4.3 ANALYSIS

**Visualization of decomposed semantic space.** As shown in Fig. 6, we visualize the semantic space with and without semantic decomposition on the AVS-Semantic dataset using t-SNE (Van der Maaten & Hinton, 2008). Three types of single-source audios ("man", "guitar", "piano") and two types of multi-source audios ("man+guitar", "man+piano") are enrolled. Without decomposition, the multi-source features are highly entangled, presenting fewer evidences related to single-source semantics. However, after performing semantic decomposition, the "man+guitar" feature presents clear evidences related to its corresponding single-source ("man" and "guitar") semantics. This is reflected in the proximity of the "man+guitar" feature to the centroids of its corresponding single-source features. The same applies to the "man+piano" feature. Note that, we omit the "background" feature in the visualization.

**Importance of the single-source audio on the semantic decomposition of multi-source audio representation.** We present empirical evidence that the single-source audio samples significantly contribute to the success of semantic decomposition. To demonstrate this, we compare the performance of our model trained on two training sets with the same number of samples: one contains solely multi-source audio samples, and the other contains single- and multi-source audio samples with the ratio of 1:1. As illustrated in Fig. 7, the model trained solely on multi-source audio samples exhibits inferior performance compared to the model trained on both types of samples, regardless of the token number and codebook size. We conjecture that the single-source samples serve as informative anchors that assist the model in learning the correct distributions of the decomposed simplex spaces for multi-source samples. In the absence of single-source samples, the decomposition task could be more difficult due to the absence of such informative anchors.

**Ablation on audio decomposition domain.** We conducted an experiment to demonstrate the benefits of conducting audio decomposition at the semantic domain instead of the time domain. Specifically, we decomposed the multi-source audio with a commonly used sound source separation model (Chen et al., 2022) and then performed audiovisual segmentation for each decomposed audio using our proposed model. The results in Table 5 clearly show that our semantic-level decomposition mechanism outperforms the time-domain decomposition approach. We attribute this improvement to two factors: 1) the imperfection of sound source separation and 2) the conflicts that arise when combining the masks for each source in the time domain without considering visual content during separation. In contrast, our semantic-domain approach does not suffer from these issues and can effectively leverage the information contained in both audio and visual modalities.

## 5 CONCLUSION

This paper presents an approach to address the challenges in audiovisual segmentation by proposing semantic decomposition of complex semantic spaces that encode multi-source audios, followed by their interaction with visual features. This reduces the semantic ambiguity in multi-source audio-visual interaction. To handle sound event changes, we propose local semantic calibration to align audio and video on a per-frame basis. Our method also incorporates a codebook sharing mechanism to enhance local audio features by distilling knowledge from that at the global level. The proposed approach is evaluated on three AVS benchmarks and the results demonstrate its superiority and effectiveness over previous state-of-the-art methods.

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

## A    MORE EXPERIMENTS

| frame number | Object-M $\mathcal{J}\&\mathcal{F}$ | Sementic mIoU |
|---|---|---|
| 3 | 60.9 | 45.4 |
| 5 | 61.6 | 46.6 |
| 7 | - | 46.6 |

Table 6: Ablation on the input frame number.

**Frame number.** We ablate the influence of input frame number during training. As shown in Table 6, we notice a frame number of five achieves the best performance. For the AVS-Object dataset, since the maximum clip length is five, we do not experiment with larger frame number. Please note that the frame number is only fixed during training and the model can accept arbitrary frame numbers during inference.

| layer number | Object-M $\mathcal{J}\&\mathcal{F}$ | Sementic mIoU |
|---|---|---|
| 1 | 61.3 | 45.6 |
| 3 | 61.6 | 46.6 |
| 5 | 61.0 | 46.0 |

Table 7: Ablation on transformer decoder layer number.

**Transformer decoder layer number.** We conduct an ablation study on transformer decoder layer numbers in semantic decoders. As shown in Table 7, a transformer decoder layer of 3 achieves the best performance. We notice that even a single-layer transformer decoder for semantic decomposition can lead to a good performance.

| frame resolution | Sementic mIoU |
|---|---|
| 224× | 46.6 |
| 640× | 49.2 |

Table 8: Ablation on input frame resolution.

**Input Resolution.** The default setting of AVSBench is $224 \times 224$ (following the sound source localization convention) for both AVS-Object and AVS-Semantic datasets. While AVS-Semantic actually provides high-resolution (720p) frames. We conduct experiments to ablate the input resolution to facilitate future comparison. Following the semantic segmentation convention, we scale the input frames to the longest side 224 or 640. The results are illustrated in Table 8. We only conduct ablation on AVS-Semantic since the resolution of AVS-Object is low-resolution ($224 \times 224$). The results are reported with the ResNet-50 backbone.

**Per-class IOU analysis.** As is shown in Fig. 8, we show the per-class iou score on the AVS-Semantic dataset. Our model demonstrates strong audio-guided segmentation capabilities for common head classes such as 'background', 'train', 'airplane', 'hair-dryer' and 'clock'. These classes are accurately segmented with a high level of precision and reliability. The model effectively distinguishes the 'background' class, providing a solid foundation for identifying and isolating foreground objects. It accurately segments transportation-related classes like 'train', 'airplane', and 'bus' capturing their intricate details and boundaries. Similarly, it excels in segmenting objects such as 'hair-dryer', 'clock' and 'tabla,' effectively separating them from the background. Even for more complex and nuanced classes like 'wolf,' our model demonstrates commendable segmentation performance, accurately delineating the contours and shape of the subject. Overall, our model showcases its ability to segment these common head classes with high accuracy and proficiency, making it a reliable choice for various segmentation tasks.

However, the scarcity of data samples for tail classes like 'utv', 'parrot', 'missile-rocket', 'harmonica', 'clipper', 'boy' and 'ax' in the presence of a long tail distribution can significantly impact the

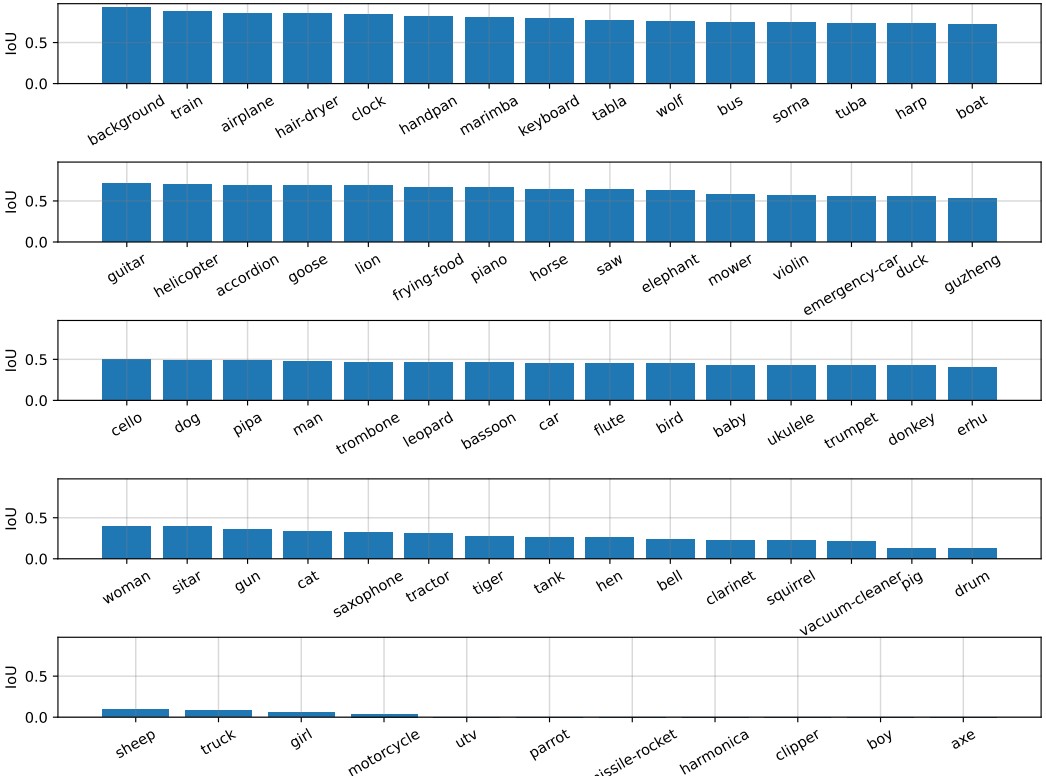

Figure 8: **Per-class IOU Analysis.** Our model demonstrates strong audio-guided segmentation capabilities for common head classes, accurately capturing 'background', 'train', 'airplane', 'hair-dryer', and 'clock' with high precision. However, the limited data samples for tail classes like 'utv', 'parrot', 'missile-rocket', 'harmonica', 'clipper', 'boy', and 'ax' due to a long tail distribution adversely affect the model's segmentation performance, hindering accurate identification and delineation of these classes.

performance of our model, specifically in the task of segmentation. With limited examples to learn from, the model finds it challenging to capture the intricate patterns and unique characteristics associated with these classes. Consequently, the accuracy and reliability of segmentation results for the tail classes may be compromised, leading to suboptimal performance in accurately identifying and delineating these objects or entities of interest.

# B    MORE RELATED WORKS

Audiovisual segmentation also closely relates to video object segmentation (VOS) Yang et al. (2018); Jain et al. (2017); Cheng et al. (2021b); Seong et al. (2020); Hu et al. (2021); Cheng et al. (2021c); Seong et al. (2021); Yang et al. (2021) and video semantic segmentation (VSS) Li et al. (2018); Sun et al. (2022); Zhuang et al. (2022); Hu et al. (2020); Paul et al. (2020). AVS requires understanding the visual contents and then corresponding them with the audio semantics to segment objects. Specifically, the most closely related task in the video segmentation domain is the referring video object segmentation (RVOS) Botach et al. (2022); Wu et al. (2022); Seo et al. (2020) which aims to segment objects in the visual frames given a linguistic expression. For each expression, RVOS only refers to one object while the AVS task permits audio query to refer to multiple objects which makes AVS task more challenging.

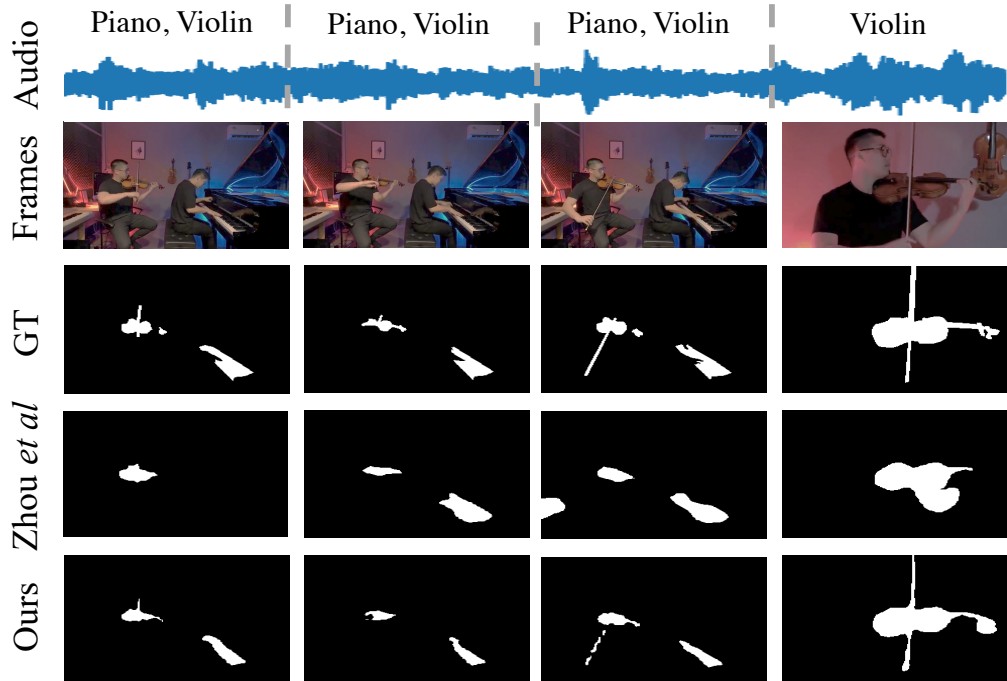

Figure 9: Qualitative comparison to Zhou et al. Zhou et al. (2023) on AVS-Object. Our method outperforms Zhou et al.'s approach by consistently and accurately segmenting the correct objects throughout the entire video clip, showcasing superior performance and better mask quality. These results emphasize the effectiveness and robustness of our approach in achieving accurate object segmentation in audio-visual scenes.

## C   MORE VISUALIZATION & VIDEO DEMO

**More qualitative results on AVS-Object.** In our study, we provide visualizations of the qualitative results on AVS-Object, as shown in Fig. 9. We compare our method with the approach proposed by Zhou et al. Zhou et al. (2023) and observe a notable difference in performance. Specifically, in the third frame of the video clip, the method proposed by Zhou et al. suffers from the false-positive problem, incorrectly segmenting objects. In contrast, our method consistently and accurately segments the correct objects throughout the entire video clip, demonstrating superior performance. Additionally, our method showcases better mask quality, with more precise and detailed segmentation boundaries. These results highlight the effectiveness and robustness of our approach in achieving accurate object segmentation in audio-visual scenes.

**More qualitative results on AVS-Semantics.** As is shown in Fig. 10, Fig. 11, Fig. 12 and Fig. 13, our model exhibits exceptional proficiency in accurately segmenting both multiple and tiny sounding objects, showcasing its versatility and robustness in audio-guided segmentation tasks. Through the implementation of a decomposed and discretized audio representation, our model effectively captures the distinct acoustic characteristics of various objects, enabling precise delineation of multiple simultaneous sound sources. Furthermore, the model demonstrates remarkable capability in capturing the intricate details and nuances of tiny sounding objects, ensuring accurate segmentation outcomes even for the smallest entities.

**Video demo (with audio).** We strongly recommend viewing the demo video provided in the supplementary materials, ensuring that you enable audio playback. Watching the video with audio will provide a comprehensive understanding of our audio-visual segmentation application, showcasing how our model utilizes a decomposed and discretized representation to achieve precise audio-visual segmentation results.

## D  MORE IMPLEMENTATION DETAILS

We set the $\lambda_{cls} = 2$, $\lambda_{L1} = 5$, $\lambda_{giou} = 2$, $\lambda_{dice} = 2$, $\lambda_{focal} = 5$, $\lambda_{com} = 0.5$ and $\lambda_{quant} = 1$ during all training process. A mask confidence threshold of 0.5 and a class confidence threshold of 0.1 is leveraged to filter out low-confident predictions. $C_v = C_e = C_q = 256$ is utilized. The positional embedding added in the transformers is the standard triangle positional embedding used in Vaswani et al. (2017). We set the layer number to three for all the transformers decoders (including local ASD, global ASD and $\mathrm{TrD}_{segm}$ in mask decoder).

### D.1  ENCODERS

**Visual encoder.** The visual encoder consists of a visual backbone and a deformable transformer encoder Zhu et al. (2020). We extract frame-level visual features from each frame $I_t$ with a shared backbone. The $T$ extracted features are then fed into the deformable transformer encoder to further conduct temporal aggregation. Let us denote the extracted visual features as $F_v = \{f_t\}_{t=1}^T$, where $f_t \in \mathbb{R}^{C_v \times H \times W}$, and $C_v$, $H$, $W$ denote the channel, height, width of the feature.

**Acoustic encoder.** We use VGGish Hershey et al. (2017) to extract audio features. Let the extracted audio feature be $F_a \in \mathbb{R}^{C_a \times L_a}$ where $C_a$ is the dimension of acoustic feature space, and $L_a$ is the audio clip length. Note that audio and video frames are already synchronized, thus the length of the audio clip is the same as the length of the video clip.

## E  MORE DETAILS ABOUT INFERENCE

To tackle scenarios where queried content keeps changing, we perform per-frame inference. For each time $t$, we assign a class to the pixel at $[h, w]$ by

$$\arg\max_{C \in \{1, \cdots, K\}} \sum_{i=1}^{N} P_{i,t}[C] M_{i,t}[h, w], \tag{11}$$

where $P_{i,t}[C]$ is the probability of class $C$. Note that $\arg\max$ does not include the "empty" category ($\emptyset$) as AVS requires each output pixel to belong to one semantic category.

| Input Video Frame | Ground-Truth | Zhou *et al* | Ours |

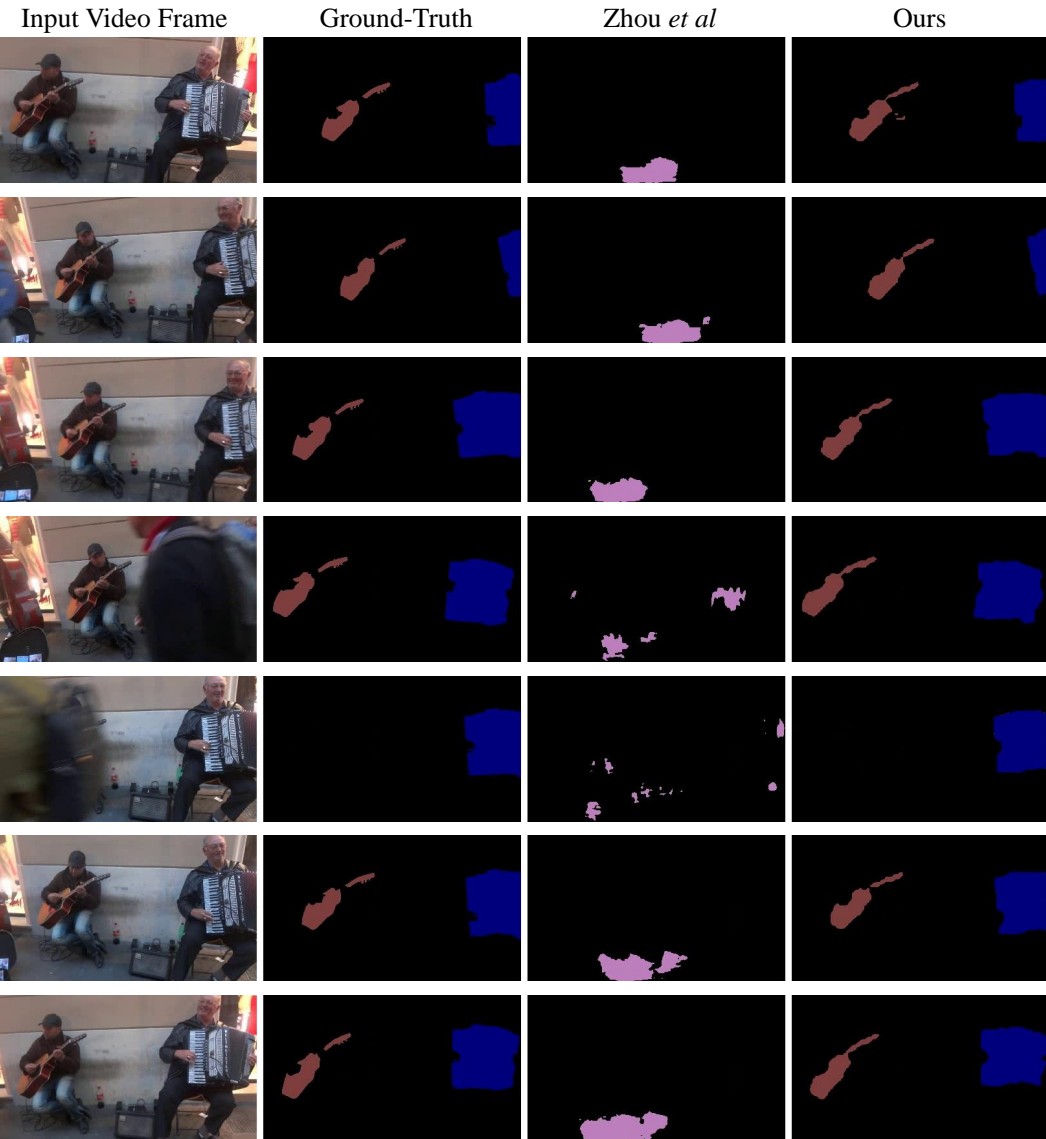

Figure 10: Qualitative comparison to Zhou et al. Zhou et al. (2023) on AVS-Semantic. Each color represents a semantic category. Our model excels in accurately segmenting **multiple sounding objects**, showcasing its proficiency in audio-guided segmentation. This success can be attributed to the effective utilization of a decomposed and discretized audio representation, which enables the model to capture and analyze the distinct acoustic features of each object, resulting in precise segmentation outcomes.

Input Video Frame      Ground-Truth      Zhou *et al*      Ours

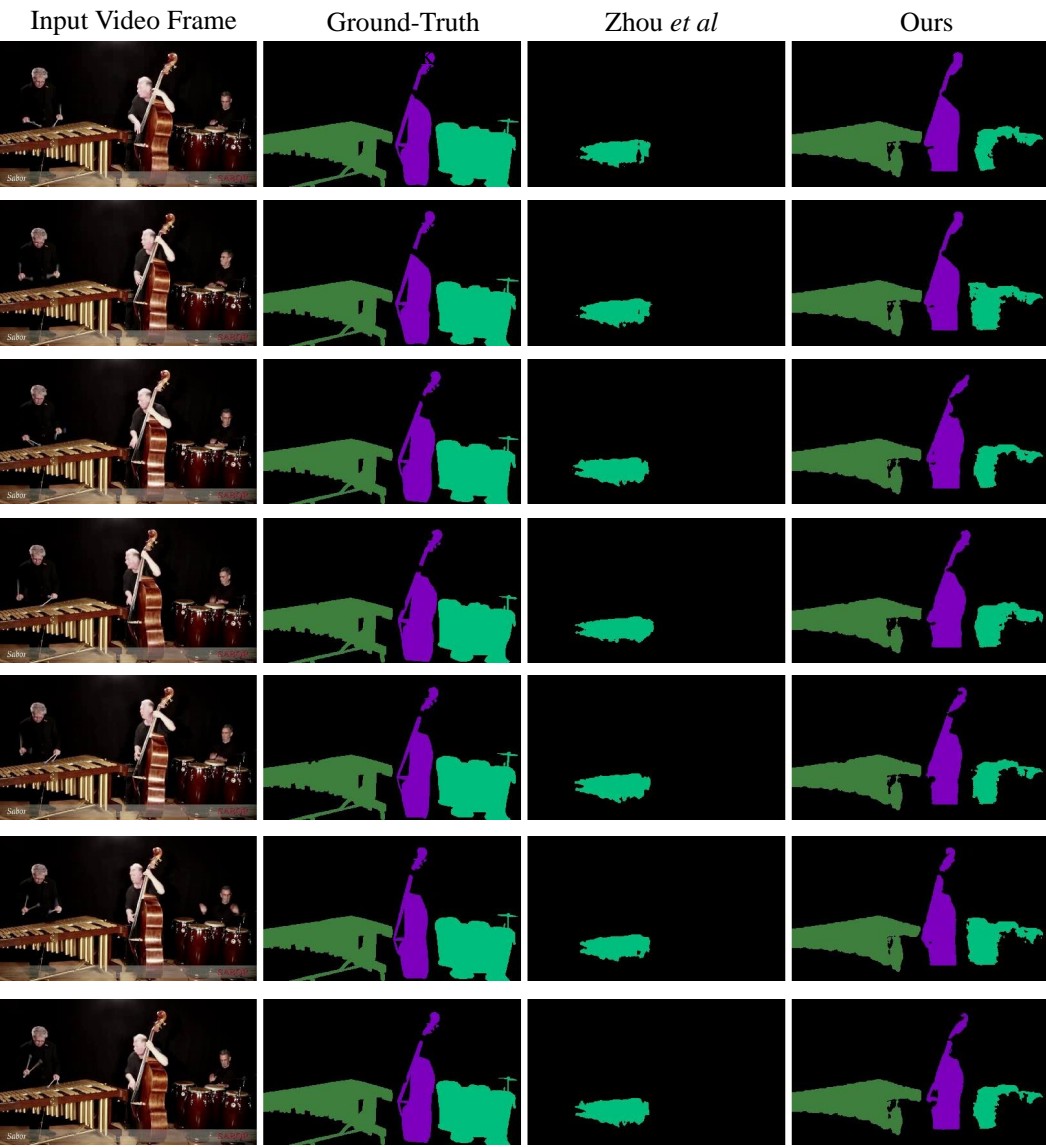

Figure 11: Qualitative comparison to Zhou et al. Zhou et al. (2023) on AVS-Semantic. Each color represents a semantic category. Our model excels in accurately segmenting **multiple sounding objects**, showcasing its proficiency in audio-guided segmentation. This success can be attributed to the effective utilization of a decomposed and discretized audio representation, which enables the model to capture and analyze the distinct acoustic features of each object, resulting in precise segmentation outcomes.

| Input Video Frame | Ground-Truth | Zhou *et al* | Ours |
| --- | --- | --- | --- |

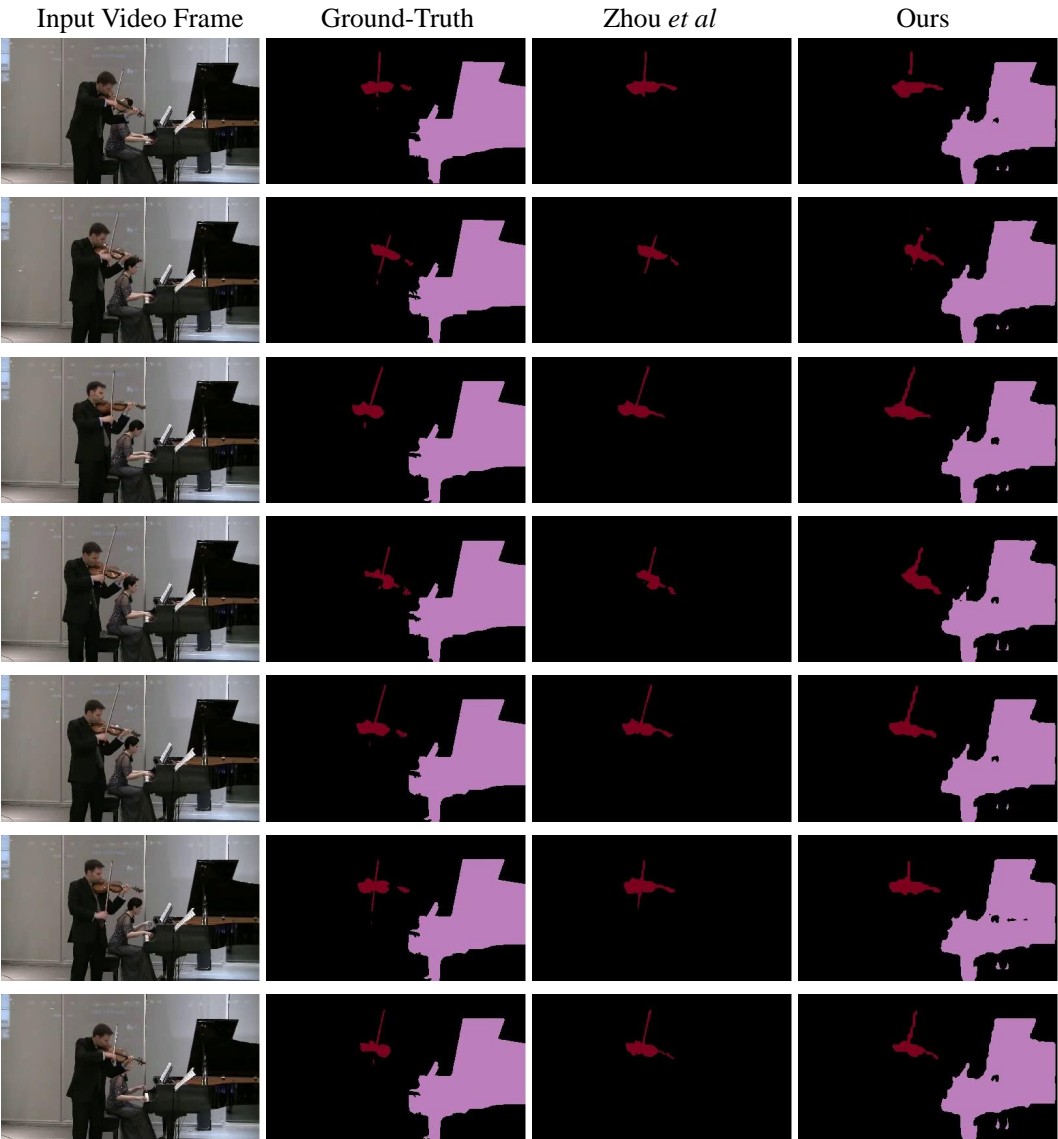

Figure 12: Qualitative comparison to Zhou et al. Zhou et al. (2023) on AVS-Semantic. Each color represents a semantic category. Our model excels in accurately segmenting **multiple sounding objects**, showcasing its proficiency in audio-guided segmentation. This success can be attributed to the effective utilization of a decomposed and discretized audio representation, which enables the model to capture and analyze the distinct acoustic features of each object, resulting in precise segmentation outcomes.

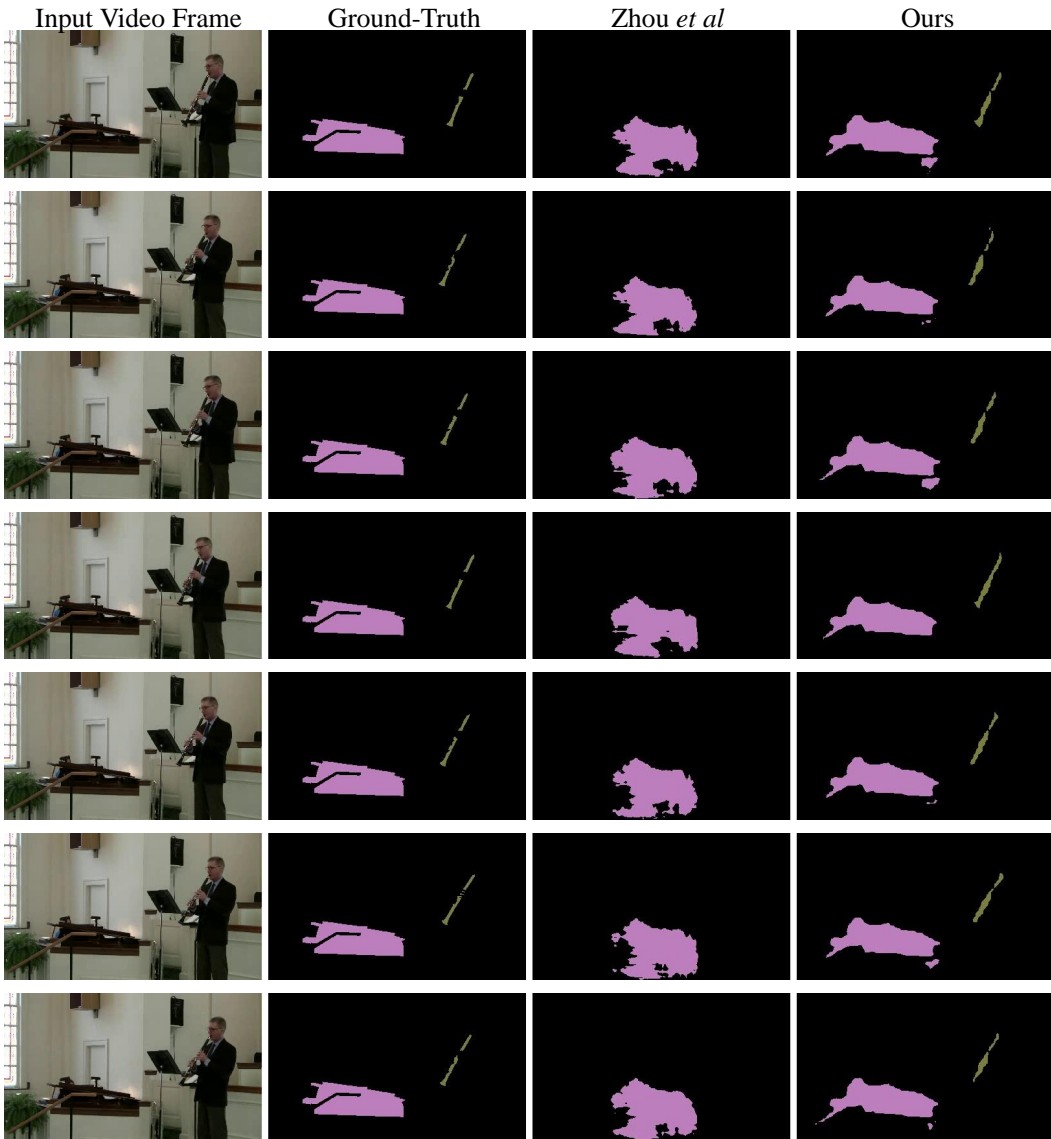

Figure 13: Qualitative comparison to Zhou et al. Zhou et al. (2023) on AVS-Semantic. Each color represents a semantic category. Our model demonstrates remarkable proficiency in accurately segmenting **tiny sounding objects**, owing to the implementation of a decomposed and discretized audio representation. By leveraging this technique, our model effectively captures the intricate acoustic details and nuances of these small-sized objects, resulting in precise and reliable segmentation outcomes.

