# OpenReview forum: "Rethinking Audiovisual Segmentation with Semantic Quantization and Decomposition"
_ICLR.cc/2024/Conference — ICLR 2024 Conference Withdrawn Submission_

### Official Review · Reviewer_1erG · 2023-10-26

**Soundness:** 3 good
**Presentation:** 3 good
**Contribution:** 2 fair
**Rating:** 5
**Confidence:** 4

**Summary:**

This paper proposes a semantic decomposition method based on product quantization, which can decompose and represent multi-source semantics by several quantized single-source semantics. Otherwise, they propose a global-to-local quantization mechanism to distill knowledge from stable global (clip-level) features into local (frame-level) ones.

**Strengths:**

1. According to the experimental results of the paper, the performance of the improved method has been significantly improved, which is a significant improvement compared to the previous methods.
2. The combination of extracting knowledge from the global level to enhance local audio features and using local semantic calibration to align audio and video is reasonable.
3. The overall writing of the paper is fluent and reasonable.

**Weaknesses:**

1. The details of the paper need to be re-checked, as shown in Figure 1 (a). It seems that the audio and video of the guitar and singing do not correspond correctly.
2. For ease of reading, some important conceptual terms need to be explained nearby, such as local ASD and global ASD in Figure 3.
3. The audio semantic decomposition via product quantization proposed by the author is mainly aimed at the scene of multi-source audio, so SQD has limited improvements over the AVS-Object dataset.

**Questions:**

1. Why did SQD improve so much when it added single-source training in figure7?
2. I am concerned that the features of multi-source audio cannot be accurately separated into the features of single-source audio. Can the author provide the visualization of the segmentation results predicted by the disentangled audio features?

---

### Official Review · Reviewer_cNHw · 2023-10-29

**Soundness:** 2 fair
**Presentation:** 3 good
**Contribution:** 2 fair
**Rating:** 5
**Confidence:** 3

**Summary:**

The paper targets the audio-visual segmentation (AVS) task that aims to segment visual objects in videos based on their associated acoustic cues. A critical point in the AVS problem is the disentanglement of the multi-source signals. Motivated by the idea that multi-source semantic space can be represented by the cartesian product of single-source. The paper designed a semantic decomposition method based on product quantization to decompose multi-source data. In addition, the paper also introduces a global-to-local quantization mechanism, which distils knowledge from stable global (clip-level) features into local (frame-level) ones, to handle the constant shift of audio semantics. Extensive experiments demonstrate the proposed method can outperform previous methods by a large margin.

**Strengths:**

The paper is generally well-written and carefully organized.

The idea of learning disentangling the representation for multi-source is interesting and important for the audio-visual learning task.

The proposed method shows better results than the baseline methods.

**Weaknesses:**

The paper did not cite all the papers published earlier this year (i.e., [1,2]). I would suggest that the author include these related works and, if necessary, make comparisons.

I believe that the core of the paper is quite similar to the prototype-learning method (i.e., [3,4]). In this method, prototypes are learned for each class during training, and during testing, similar metrics are used to determine the highest similarity of the input feature with respect to the prototypes. However, the paper lacks a discussion of the use of such a method in the literature review, and the novelty of the PQ-based multi-source semantic decomposition is somewhat overstated. Additionally, instead of the 'decomposition' of sound sources into a class-specific manner, the proposed method attempts to use these prototypes in a class-agnostic manner, which might hinder the visualization and understanding of the learned prototypes.

I am not sure about the "|" notation in Eq.6, does this equation mean we have two inputs for the network?

In Tab.1 of the experimental section, it is good to see that the author compares different backbones for the evaluation. However, it is unclear to me why the author decided to use Swin-Tiny and V-Swin-Tiny to compare with the PVT-v2-Base baseline TPAVI (Zhou et al., 2023). It looks like an unfair comparison in this case. Additionally, it looks strange to me that the proposed method only improved by 0.8% under AVS-Object-Single (Jaccard) but improved by 23.6% on the AVS-Semantic dataset (mIoU).

The proposed framework exhibits certain similarities with ReferFormer [5]. However, the paper does not include the results of ReferFormer, which could be placed in either Tab.1 or Tab.2.



**Minors**
In Figure 1 the Guitar and Singing are misplaced from each other.



**Reference**
[1] Li, K., Yang, Z., Chen, L., Yang, Y. and Xun, J., 2023. Catr: Combinatorial-dependence audio-queried transformer for audio-visual video segmentation. arXiv preprint arXiv:2309.09709.
[2] Chen, Y., Liu, Y., Wang, H., Liu, F., Wang, C. and Carneiro, G., 2023. A Closer Look at Audio-Visual Semantic Segmentation. arXiv e-prints, pp.arXiv-2304.
[3] Snell, J., Swersky, K. and Zemel, R., 2017. Prototypical networks for few-shot learning. Advances in neural information processing systems, 30.
[4] Chen, C., Li, O., Tao, D., Barnett, A., Rudin, C. and Su, J.K., 2019. This looks like that: deep learning for interpretable image recognition. Advances in neural information processing systems, 32.
[5] Wu, J., Jiang, Y., Sun, P., Yuan, Z. and Luo, P., 2022. Language as queries for referring video object segmentation. In Proceedings of the IEEE/CVF Conference on Computer Vision and Pattern Recognition (pp. 4974-4984).

**Questions:**

Please refer to the weakness section above.

---

### Official Review · Reviewer_vB2P · 2023-11-08

**Soundness:** 3 good
**Presentation:** 2 fair
**Contribution:** 2 fair
**Rating:** 5
**Confidence:** 3

**Summary:**

This paper handles the tasks of audio-visual segmentation and audio-visual semantic segmentation, where the goal is to predict precise masks for the objects that make sound in the video. For the semantic segmentation task, the model must also predict the correct object class for the mask. This paper proposes a model to improve these tasks using vector-quantized codebooks and a global-to-local approach of distillation into the codebooks. Models are trained and evaluated on the AVS-Object dataset and AVS-Semantic datasets where state-of-the-art performance is reported.

**Strengths:**

- A new model to fuse audio and visual features is proposed which incorporates vector quantization.
- The high-level idea of the global to local approach using a shared VQ codebook is intuitive.
- State of the art performance is reported on both benchmarks.
- Many useful ablation and analysis studies are provided as well as qualitative results.

**Weaknesses:**

- The proposed method is just a pipeline of existing methods (transformer decoder, VQ, and feature modulation). Vector quantization for cross-model learning has been proposed in [1] and global-local audio-visual representations have been explored in [2]. Also, the proposed method is relying on pre-extracted features (if I understand correctly, the visual and audio feature backbones are not trained).
- The contribution of this work is narrow - a specialized architecture is proposed only for audio-visual semantic segmentation. I think this paper would be stronger if the method was evaluated on a large-scale dataset like AudioSet and VGGSound (similar to the evaluation in [2]) or if the method was evaluated on different modalities like [1] does (ie. image-text or speech-image).
- The paper claims an improvement over SOTA of 25.4 -> 46.6 mIoU for the AVS-Semantic task, while the authors' baseline model in Table 2 without the proposed modules already achieves 33.5 mIoU, so the impact is less than stated. Also, this doesn't report results for the F-Score on the AVS-Semantic task which was reported in prior results.
- For the AVS-Object task, the proposed method mainly improves on the mIoU (region similarity) and the improvement on contour accuracy F is less significant.
- The writing is unclear in several sections and there are distracting typos ("Sementic"). I did not understand part of the methods section. Figure 1 seems incorrect - the person is highlighted for "guitar" and the guitar is highlighted for "singing."

[1] Alexander Liu, SouYoung Jin, Cheng-I Lai, Andrew Rouditchenko, Aude Oliva, and James Glass. 2022. Cross-Modal Discrete Representation Learning. In Proceedings of the 60th Annual Meeting of the Association for Computational Linguistics (Volume 1: Long Papers), pages 3013–3035, Dublin, Ireland.

[2] Zeng, Zhaoyang, Daniel McDuff, and Yale Song. "Contrastive learning of global and local video representations." Advances in Neural Information Processing Systems 34 (2021): 7025-7040.

**Questions:**

Method
- I didn't understand how the global audio semantic decoder (global ASD) + the set of learnable semantic prototypes works. What are the set of learnable semantic prototypes? What is the input and output into the transformer? Why is it a decoder (is there something autoregressive about the input / output)?
- What is the input to the local semantic decoder? Is it the audio feature corresponding to a single point in time? In Figure 3, I don't understand how the output of the local semantic decoder is a sequence while the input is only a single audio feature.

Results
- I'm not sure if this work makes a fair comparison with prior work since it incorporates a "deformable transformer encoder" after the frame-level visual features, and this extra transformer does not seem to be used in prior work. How many parameters does this add, and how does your model perform without it?
- How does your model compare in terms of design / architecture to the prior audio-visual fusion methods for segmentation? How many parameters do your proposed modules have versus theirs?
- How does your baseline model in Table 2 without the proposed modules already achieve better mIoU than SOTA (25.4 from AOT -> 33.35 with your baseline)?
- For the AVS-Object task, the proposed method significantly improves the mIoU (region similarity) but the improvements for contour accuracy F are much smaller - why is this?
- For AVS-Semantic, Zhou et. al 2023 also reports the F-Score besides mIoU. How does your method perform with this metric?
- For the loss functions, the prior work Zhou et. al 2023 does not use a bounding box loss and classification loss. How much do these losses impact the performance? Also, they directly use the features from the decoder with a softmax operation to predict the class, while your method uses a two layer fully connected network. How does this difference impact the performance?
- What is the explanation for the huge improvement in mIoU for the AVS-Semantic task? Could you possibly show a per-class IOU analysis for the baseline model to see the differences?
- I didn't understand the analysis "Importance of the single-source audio on the semantic decomposition of multi-source audio representation." Do you train on a mix of the AVS-object and AVS-Semantic datasets during regular training, and you are trying to show why this is beneficial?
- How does the model handle two instances of the same class in the video? For example, if two people playing violin are in the same video, but only one of them is playing.